# The Role of Chemerin in Neutrophil Activation and Diseases of the Lung

**DOI:** 10.3390/biomedicines13061354

**Published:** 2025-05-31

**Authors:** Patrick Arndt

**Affiliations:** Division of Pulmonary, Allergy, Critical Care and Sleep Medicine, University of Minnesota, Minneapolis, MN 55455, USA; arndt108@umn.edu

**Keywords:** chemerin, neutrophil, lung, cancer, COPD

## Abstract

Chemerin is an adipokine with complex biochemistry that undergoes proteolytic modification by components of the inflammatory, coagulation, and fibrinolytic systems, generating both active and inactive products. Chemerin has been found to have both pro- and anti-inflammatory properties, can regulate angiogenesis, and is involved in cancer pathogenesis. Although chemerin is a chemoattractant for macrophages, plasmacytoid dendritic cells, and natural killer cells, it does not induce neutrophil chemotaxis. In contrast, neutrophils appear to act on chemerin structure and localization to regulate the inflammatory response. A role for chemerin in several lung diseases, including airway disease, interstitial lung disease, and cancer, has begun to be explored, but its full role is yet to be fully understood. This review will discuss the role of chemerin in neutrophil activation and will examine what is currently known of the effect of chemerin in diseases of the lung.

## 1. Introduction

Chemerin is an adipokine that was initially identified by Wittamer et al. in 2003 as an agonist for the then-orphan receptor ChemR23 [1]. Based on the structure of ChemR23, it was predicted to function as a receptor for chemoattractants for antigen-presenting cells and is highly expressed on macrophages, natural killer cells, and plasmacytoid dendritic cells [1]. Chemerin was originally identified in ascitic fluid from patients with ovarian and liver cancer, synovial fluid from arthritic individuals, and splenic extracts, thereby strengthening its proposed role in cancer and inflammation. Based on this, much enthusiasm was generated regarding a potential role of chemerin in regulating the inflammatory response to trauma or infection but also in the natural immune response to tumors. Since its initial description, chemerin has been shown to be expressed by and released from several tissues and cells, including the liver, lung, spleen, lymph nodes, epithelium, endothelium, fibroblasts, macrophages, and dendritic cells, suggesting a wide involvement in several disease states [1,2].

Since its discovery, much has been learned regarding the biochemistry of chemerin, particularly of the proteolytic steps needed for its activation. However, its role in regulating inflammation after injury or infection, as well as its role in tumor growth and spread, is only beginning to be understood. Barriers to fully understanding the role of chemerin in specific diseases are interpreting the importance and role of systemic versus localized chemerin levels, the activity of individual proteolytic products, and ongoing proteolysis, as well as the effect of chemerin binding to other non-ChemR23 receptors [3]. This has been highlighted in several studies, as will be discussed below, as chemerin has been shown to be both pro- and anti-inflammatory depending on its localization, the timing of tissue influx, proteolytic cleavage (and subsequent proteolysis), and specific cellular bonding. These complexities are not fully appreciated but are being actively investigated in several animal models and in human clinical samples of disease. Thus, much is still to be learned of the chemistry of chemerin and its role in human disease prior to decisions regarding potentially targeting chemerin or its receptors in the treatment of disease. The goals of this review are two-fold and include, firstly, an examination of the role of chemerin in regulating neutrophil activation and their functional responses and the role of neutrophils in the biochemistry and response to chemerin in vivo and, secondly, the current understanding of the role of chemerin in the pathophysiology of individual lung diseases. This review will highlight that much is known regarding the biochemistry of chemerin but its role in the onset, progression, and resolution of lung disease, as well as the local biochemistry in vivo, remains poorly understood.

## 2. Biochemistry of Chemerin

Chemerin was initially isolated as the agonist for the orphan ChemR23 from several human biologic samples, including the ascitic fluid of patients with cancer and synovial fluids from patients with inflammatory arthritis, and is widely expressed by several cells and tissues [1]. Chemerin is abundant in plasma and circulates in a prochemerin form that needs to undergo proteolytic cleavage to induce activity, with this cleavage occurring extracellularly [3,4,5,6,7]. An interesting feature of chemerin is its proteolytic cleavage profile that leads to fragments with variable biological activity or to fragments that lack any biological activity, with subsequent proteolysis occurring locally during an inflammatory response [3,4,5,6,7]. This can thereby alter the activity and binding of chemerin locally, to induce either a pro- or anti-inflammatory response, and therefore its role in disease. Several cleavage sites are present on chemerin with unique proteases targeting these sites, resulting in the several proteolytic products that have been described. Broadly, chemerin has been shown to undergo proteolysis by components of the inflammatory, coagulation, and fibrinolytic cascades. A full understanding of the kinetics and localization of this protein processing is presently unavailable but will likely be key to unraveling the role in human disease, including in acute inflammation and in the resolution of the acute inflammatory response, as well as in tumor biology.

Originally, prochemerin was found to be cleaved by serine proteases derived from mature neutrophils including elastase and cathepsin G, producing the active chemerin products chem^21−157^ and chem^21−156^, respectively, with chem^21−156^ being slightly less active of the two [5] (Figure 1). In regard to elastase, two other cleavage sites have been identified on chemerin, producing chem^21−155^ and chem^21−152^, each with variable activity, again highlighting the complexity of chemerin and its study in human diseases. A later study by Zabel et al. expanded the proteases able to cleave chemerin to include plasmin, urokinase plasminogen activator, tissue plasminogen activator, factor VIIa, factor XIIa, and mast cell-derived tryptase [4]. Although the study by Zabel et al. did not identify factor X1a as a protease for prochemerin, subsequently, Ge et al. was able to observe that factor XIa can also cleave chemerin into an active form, thereby expanding the role of the coagulation system in regulating chemerin activity [8]. The carboxypeptidases N and B have also been shown to cleave chemerin into biologically active fragments with chemoattractant activity [6]. In contrast to the proteases described above, carboxypeptidases B and N are unable to cleave full-length chemerin but are able to cleave a carboxy-terminal amino acid of plasmin-cleaved prochemerin, generating a product with higher chemoattractant activity than that generated by plasmin cleavage alone [6]. This highlights the importance of sequential chemerin cleavage in its localized biological activity. The above discussion of prochemerin and chemerin processing focused on the effect of endogenous proteases. Emphasizing the role of chemerin in inflammation induced by infection is the interesting observation by Kulig et al. that an exogenous protease, staphopain B, released from Staphylococcus aureus, was able to cleave chemerin into a biologically active mediator able to induce chemotaxis [9]. This is the only study identified to date that demonstrates that bacterial species produce proteases targeting chemerin and highlights the importance of chemerin during the infectious process. Undoubtedly, other bacterial-, viral-, or fungal-derived proteases will be identified that regulate chemerin activity and that could be specific targets for the design of therapeutic treatments.

In contrast to the above description of proteases that generate active forms of chemerin, two enzymes, neutrophil derive proteinase 3 and mast cell chymase, were shown to generate chemerin products that lacked biological activity [10]. This process is seen as anti-inflammatory and may represent one way in which inflammation is regulated after the initial acute inflammatory response with recruited neutrophils responsible for tampering with inflammation through ongoing chemerin proteolysis. Unique differences exist in the effects of proteinase 3 and mast cell chymase on the cleavage of chemerin. Whereas proteinase 3 is active on prochemerin and not its active form (chem^21−157^), mast cell chymase only cleaves an active form of chemerin, and not prechemerin, into a form without activity [10]. Finally, in an attempt to further characterize the anti-inflammatory features of chemerin, Cash et al. identified that a 15-amino-acid fragment of chemerin (chemerin15 (C15), chem^140−154^) had anti-inflammatory properties and could decrease inflammatory cytokine release from macrophages and inhibit neutrophil and monocyte recruitment in a murine model of peritonitis [11]. Taken together, the processing of prochemerin into active or inactive constituents is regulated during infectious/inflammatory insults but also during hemorrhage or the activation of the coagulation cascade. Further studies are necessary to examine the localized levels of chemerin cleavage products, and their kinetics of production, to better understand the role of chemerin in inflammation and tumor biology.

A further complexity in understanding the role of chemerin in regulating inflammation is its ability to signal through additional receptors apart from ChemR23, including binding to a non-functional receptor that does not induce an intracellular response upon binding. Additional identified receptors for chemerin include GPR1 and C-C motif chemokine receptor-like 2 (CCRL2) [3,12,13]. GPR1 has been described as having a high affinity for chemerin but only weak intracellular signaling, as assessed by calcium mobilization and Erk1/2 activation [3]. This receptor has been suggested to function as a decoy receptor for chemerin, although its full functional activity remains unknown. The receptor CCRL2 has been shown to be expressed on neutrophils and may function as a receptor that facilitates chemerin localization to areas of acute inflammation or ongoing repair, as it does not induce intracellular activation [3,13]. Furthermore, CCRL2 was shown to be anti-inflammatory in a murine model of zymosan-induced peritonitis [14]. Finally, Ji et al. showed that CCRL2 expression acting on neutrophils induced neutrophil reverse migration, providing a physiologic mechanism for this anti-inflammatory event [15].

## 3. Effects of Chemerin on Neutrophil Activation

Neutrophils play an important role in the innate immune system during the acute inflammatory response through their release from granules and proteolytic enzymes, the generation and release of reactive oxygen species, the expression of cytokines and chemokines, and the release of neutrophil extracellular traps, all designed to target infectious agents and respond to cellular injury. Neutrophils are also important in the recruitment of other inflammatory cells, particularly monocytes/macrophages and lymphocytes, in mediating the adaptive immune response. Unlike plasmacytoid dendritic cells, natural killer cells, and macrophages, chemerin does not induce neutrophil chemotaxis [1]. In addition, there have been no reports of chemerin inducing or regulating the release of constituents of cytoplasmic granules, cytokines/chemokines, or neutrophil extracellular traps or playing a role in the generation of reactive oxygen species. As described above, neutrophils participate in the proteolytic cleavage of chemerin, generating inactive products [5,10]. Interestingly, this was shown to only occur after neutrophil degranulation, suggesting that it is a later process in the inflammatory cascade, a stage leading to the downregulation of inflammation and the onset of resolution and repair [5]. Although neutrophils were not shown to express ChemR23 at baseline, Cash et al. showed that ChemR23 expression was upregulated in neutrophils upon stimulation, with preformed ChemR23 present in neutrophil granules [16]. Exposure to the chemerin peptide C15 activated neutrophil-expressed ChemR23 and decreased neutrophil adhesion and transmigration [16]. As described above, neutrophils do not express ChemR23 at baseline but have been shown to express CCRL2, which may regulate neutrophil reverse migration [3,13,15]. In patients with rheumatoid arthritis, CCRL2 was shown to be upregulated in synovial fluid neutrophils, compared to circulating neutrophils from healthy controls, suggesting its role in this disease process [17]. In addition to its binding of chemerin and the physiologic presentation of chemerin to ChemR23-expressing cells, CCRL2 has also been shown to facilitate neutrophil chemotaxis via CXCR2 by forming heterodimers with that receptor [18].

## 4. Chemerin as a Regulator in Lung Infection

A few studies have begun to explore the role of chemerin in acute pulmonary infections, but its role in these models remains poorly understood. In a model using the pneumonia virus in mice, signaling through ChemR23, and presumably an effect of chemerin, was shown to be protective as ChemR23-deficient mice had accentuated neutrophil recruitment to the lung with an associated decrease and delay in viral clearance, worsening clinical signs of infection, more restrictive lung volumes, and higher mortality rates. Mechanistically, this was thought to be mediated via a decrease in type-1 interferon levels and a decrease in the recruitment of plasmacytoid dendritic cells [19]. More importantly, however, it was the loss of an anti-inflammatory response in ChemR23-deficient mice that was thought to be the more significant mechanism for the worsening of clinical signs and mortality in this model.

No study to date has directly examined the role of chemerin or its receptor ChemR23 in a bacterial model of lung infection. However, one study has examined the role of chemerin after airway lipopolysaccharide (LPS) exposure [20]. As LPS is a cell wall constituent of Gram-negative bacteria, findings from the study may suggest a role for chemerin in Gram-negative bacterial pneumonia. In the study, the administration of chemerin at the time of LPS exposure led to an overall decrease (up to 70%) of neutrophil recruitment to the lung with a delay in the time to peak recruitment. This effect was shown to be mediated via ChemR23 as the effects of chemerin on lung neutrophil recruitment were no longer present in ChemR23-deficient mice. Inflammatory mediators in BAL fluid were also reduced by chemerin treatment, including the cytokines interleukin 1, interleukin 6, and tumor necrosis factor alpha, as well as the neutrophil chemoattractant KC. Taken together, these findings suggest that chemerin may play an anti-inflammatory role in regulating the innate immune response upon exposure to bacterial cell wall constituents. However, similarly to in the murine viral pneumonia model described above, it is possible that hemerin may delay bacterial clearance in the lung in the setting of pneumonia, due to the decrease in neutrophil recruitment and cytokine expression, thereby increasing clinical severity and worsening outcomes. This possibility has not been examined to date.

From a clinical infection standpoint, serum chemerin levels have been examined in patients with sepsis with an examination of their relevance to clinical outcomes. Serum chemerin levels were increased within 48 h in a cohort of patients with sepsis in an intensive care unit (ICU) compared to healthy controls. Over the subsequent week, chemerin levels decreased in the sepsis cohort but remained elevated compared to healthy controls [21]. Importantly, serum chemerin levels correlated with the severity of sepsis, being higher in those with septic shock, and also correlated with mortality. Importantly, the results from the study may be applicable to a broad etiology of sepsis, and not just to Gram-negative infections and LPS exposure, as 23% of the included patients with identified infections had Gram-positive bacteria and 16% had fungal infection, although a subgroup analysis was not performed by the study’s authors. Chemerin levels in the study also correlated with the validated sepsis severity scores of acute physiology and chronic health evaluation (APACHE) II and the sequential organ failure assessment (SOFA) and were an independent predictor of 28-day mortality when adjusted for APACHE II scores. Interestingly, however, chemerin levels did not correlate with levels of circulating pro-inflammatory cytokines in these patients. Horn et al. showed similar results of an increase in chemerin levels in a selective study of patients with abdominal sepsis, with these findings replicated in an animal model of intraperitoneal sepsis [22]. The effects of the serum chemerin level on mortality in the study depended on the clinical presence of stress hyperglycemia in the patients with sepsis. Finally, in contrast to the above studies, Amend et al. did not detect a difference in serum chemerin levels in patients with sepsis as a whole and observed no correlation of chemerin levels with the use of vasopressors or mortality [23]. In the study, however, subgroup analysis did show a correlation of an increase in chemerin levels in those with Gram-positive bacterial infections, although this was only for a small subset (10%) of the patients with sepsis included in the study as a whole. Taken together, these studies suggest that chemerin may be a biomarker for sepsis and septic shock that can be utilized to guide therapy and provide prognosis. Further studies are necessary prior to using it in a clinical context.

## 5. Role of Chemerin in Airway Diseases—Asthma and Chronic Obstructive Pulmonary Disease

In both asthma and chronic obstructive pulmonary disease (COPD), a role for chemerin has been examined. As the pathophysiology of both diseases involves the recruitment of inflammatory cells (neutrophil, monocytes, eosinophils, and dendritic cells) to the airway epithelium and alveolar space, as well as the upregulation of inflammatory cytokines and chemokines, with both of these processes mediated by chemerin, examining a role for chemerin in these diseases is justified. As will be discussed below, although the role of chemerin is only beginning to be evaluated in these diseases, results have been inconsistent. This may be due to the model of asthma or COPD examined or other clinical confounders of the patients examined including metabolic profile and obesity, as determined by body mass index (BMI). These confounders can influence the baseline plasma levels of chemerin.

In COPD, several smaller studies have examined the role of chemerin in establishing the diagnosis, as a marker of clinical status, and as a predictor of prognosis in patients with COPD. In a study by Boyuk et al., baseline serum chemerin levels were increased in patients with COPD compared to healthy controls [24]. However, there were no findings of an increase in chemerin levels across worsening stages of COPD, with those with stage 2 having the highest levels compared to stages 1 and 3 [24]. One criticism of the study is that the results were not corrected for BMI, which was higher in the patients with COPD, with obesity known to influence chemerin levels [24]. One additional interesting finding from the study was that chemerin levels were increased in smokers compared to non-smokers, even in those who had discontinued smoking in the distant past. This suggests that airway tobacco exposure may upregulate the production and/or release of chemerin, or that the acute inflammatory response associated with smoking increases the cleavage of prochemerin to chemerin, and that these effects are longstanding. Therefore, a chronic increase in serum chemerin levels (and also, possibly, in the lung as well) may propagate the ongoing acute inflammatory response induced by smoking and explain the persistent decline in lung function in patients with COPD even after the discontinuation of smoking. In contrast to the above study, Goktepe et al. and Galecka et al. did not find a difference in the circulating levels of chemerin between patients with COPD and healthy controls, even when they stratified by COPD stage or BMI [25,26]. Other studies have suggested that chemerin would be useful in the management of COPD exacerbations or to assess the response to treatment. Li et al. showed that chemerin levels were increased in patients with COPD hospitalized with acute exacerbations and correlated with the frequency of hospitalizations and the levels of acute inflammation as determined by C-reactive protein (CRP) [27,28]. Similar findings were observed by Fu et al., wherein serum chemerin levels were increased during COPD exacerbations but subsequently declined once the exacerbation resolved [29]. Accordingly, chemerin may be useful to guide the response to treatment during a flare of COPD. Finally, one study has shown that treatment with an inhaler utilizing both a corticosteroid and a long-acting beta agonist bronchodilator decreased serum chemerin levels compared to levels prior to the initiation of treatment [30]. These results also correlated with the levels of interleukin 8 and CRP, thereby suggesting a mechanism for how chemerin may reduce airway inflammation in COPD. Several other studies have explored mechanisms for a potential role of chemerin in COPD utilizing animal models. Cigarette smoke and environmental pollutants are known triggers for the development of COPD. In examining the chemerin axis in cigarette-induced COPD, Demoor et al. observed that after both acute and chronic exposure to cigarette smoke, chemerin levels in the alveolar space, as assessed by bronchoalveolar lavage fluid, were increased compared to control animals [31]. In contrast, levels in the lung epithelium were decreased. This suggests that chemerin is released from the lung epithelium during acute inflammation but that ongoing gene expression is decreased. To further explore the role of chemerin signaling in their model of cigarette smoke-induced COPD, ChemR23 (CKMLR1)-deficient mice were exposed to cigarette smoke and were compared to wild-type cigarette-exposed mice. The results showed that ChemR23-deficient mice had decreased inflammation, as assessed by cell recruitment and chemokine levels, compared to the wild-type mice. However, neither lung physiology via plethysmography nor histologic findings of emphysema were examined in the study. In addition, the effect of chemerin and its receptor ChemR23 on the resolution of lung inflammation after cigarette exposure could not be fully examined. Similar results to the above were seen after exposure to diesel exhaust particulate (DEP) matter [32]. Similarly to cigarette smoke exposure, chemerin levels in BAL fluid were increased after DEP exposure, whereas levels in the lung epithelium were decreased. Again, acute cellular influx and chemokine levels were also decreased in ChemR23-deficient mice after exposure to DEP compared to wild-type controls.

Several studies have also examined the role of chemerin in asthma. Plasma chemerin levels were observed to be significantly increased in patients with severe persistent asthma compared to healthy controls, with an increasing trend in levels seen in less severe forms [33]. Chemerin levels also correlated with the number of Th9 and Th17 cells in circulation, which are known to be involved in the pathophysiology of asthma. Further evidence that chemerin mediates the airway response in asthma comes from a study by Zhao et al. wherein ovalbumin-sensitized mice were administered chemerin intranasally prior to exposure to ovalbumin [34]. The co-administration of chemerin decreased T-cell and eosinophil recruitment to the alveolar space, decreased peribronchiolar cellular infiltrate, and was associated with a decline in the Th-2 cytokines IL-4 and IL-13. In addition, there was a physiologic effect of chemerin administration, as shown by the decrease in airway hyperactivity by methacholine challenge in mice treated with chemerin. Mechanistically, these effects were thought to occur through CCL2, as levels of CCL2 were decreased in animals treated with chemerin and the administration of exogenous CCL2 in their model reversed these effects [34]. To strengthen their findings of the role of chemerin in allergic asthma, these authors showed similar effects of chemerin in a separate asthma model using house dust mite (HDM) exposure. Supportive results for the role of chemerin in asthma were seen by Doyle et al. when administering an engineered long-acting chemerin analog in an ovalbumin-allergic airway model [35]. Similar to Zhao et al., they observed a decrease in cellular influx to the lung, a decrease in mucus production, and less bronchial epithelial cell activation. Importantly, the administration of chemerin alone did not induce any cellular infiltrate or epithelial changes in the absence of airway challenge. In further examining a role of chemerin in asthma and its signaling through ChemR23, using a murine asthma model of combined exposure to DEP and HDM, Provoost et al. showed that BAL fluid chemerin levels significantly increased after the combined exposure compared to control animals or those exposed only to DEP or HDM [32]. However, in contrast to the earlier findings of cellular lung recruitment after DEP exposure described above, mice deficient in ChemR23 expression showed an increase in cellular influx (including neutrophils, monocytes, eosinophils, and T cells) after combined exposure to DEP and HDM compared to wild-type controls [32]. Finally, ozone is a known airway irritant, and exposure can lead to hyperactive airways and an asthma phenotype. Exposure to ozone increased chemerin levels in BAL fluid, suggesting a role of chemerin signaling in ozone-mediated disease [36]. Furthermore, the absence of chemerin, achieved using chemerin-deficient mice, resulted in a decrease in the levels of the receptor for advanced glycation end-products (RAGE) and osteopontin after ozone exposure compared to wild-type control mice, although chemokine levels did not differ in the study. Further exploration of the role of chemerin signaling, utilizing ChemR23-deficient mice, in ozone-mediated lung disease has not yet been undertaken.

## 6. Elucidation of the Role of Chemerin in Lung Cancer

As highlighted earlier, chemerin was initially isolated from ascitic fluid from patients with liver and ovarian cancer, suggesting a possible role of chemerin in the pathophysiology of cancer [1]. Through its multiple roles in metabolism, cellular recruitment, and migration, as well as angiogenesis, chemerin has several potential pathways by which to regulate tumor growth, metabolism, and metastasis [37,38]. Goralski et al., in their review, have outlined the role of chemerin in several different tumor types as well as the proposed mechanism of its effect through cellular recruitment, the regulation of cellular signaling pathways, the expression of inflammatory mediators, and effects on angiogenesis [38]. The point stressed is that the role of individual chemerin isoforms, their interaction with known chemerin receptors, and the effect of localized (tumor) versus systemic expression are at present unknown but will be important areas of future investigation to improve our understanding of the role of chemerin in cancer [38]. As it relates specifically to lung cancer, several studies have examined the expression level and potential clinical and prognostic significance of chemerin in non-small lung cancer (NSCLC) with conflicting results. In a study by Xu et al., levels of serum chemerin were examined as a potential diagnostic and prognostic marker for NSCLC [39]. The study was an extension of the findings by Qu et al., which showed that serum chemerin levels were increased in patients with NSCLC compared to healthy controls [40]. On comparing serum chemerin levels in 189 patients with NSCLC compared to healthy controls, Xu et al. showed that serum chemerin levels were increased in patients with NSCLC compared to controls, with a level of chemerin over 1500 ng/mL being able to be used as a discriminating value [39]. In addition, in patients with NSCLC, chemerin levels correlated with tumor stage and distant metastasis and were inversely correlated with overall and progression-free survival, with those with higher chemerin levels having a decreased life expectancy [39]. Although Qu et al. did not find a correlation of chemerin levels with clinicopathologic grade, their study population was significantly smaller than that of Xu et al. Similar findings to those above were observed by Sotiropoulos et al. in patients with resectable NSCLC, with, again, a correlation of serum chemerin levels with tumor stage [41]. In contrast to the above studies, Goktepe et al. observed the opposite effect wherein there was no difference in serum chemerin levels between patients with lung cancer and healthy controls [25]. Theirs was a small study, and the wide standard deviation in chemerin levels observed in the patients with lung cancer may have potentially obscured their results, as chemerin levels tended to be higher in patients with lung cancer compared to controls. In addition, age matching was not present in the healthy controls compared to patients with lung cancer, with age known to correlate with serum chemerin levels. Overall, taken together, these studies suggest that serum chemerin may be both a diagnostic and prognostic tool in patients with NSCLC. These findings are in contrast to those from studies examining chemerin levels in lung tumors. Cai et al. examined tazarotene-induced gene 2 (TIG2, chemerin) mRNA and protein levels, by Western blotting and immunohistochemistry, in surgically resected tumors from patients with NSCLC compared to the surrounding normal tissue [42]. In contrast to the findings for serum chemerin, a decrease in the tumor expression of chemerin was found to correlate with the stage of cancer, tumor differentiation, and overall survival by multivariate analysis [42]. One caveat of the study was that there were some differences in the results of protein expression between Western blotting and immunohistochemistry (IHC) techniques, with a less significant difference in chemerin levels as detected by IHC. Similar results were also obtained by Zhao et al., wherein almost 52% of patients with NSCLC showed a decrease in tumor tissue chemerin expression by IHC (similar to the 58% seen by Cai et al.), with a decrease in levels correlating with histologic grade and a decrease in survival [43]. Mechanistically, they proposed that decreased tumor expression of chemerin results in a decrease in tumor immunogenicity as they also observed that chemerin levels correlated with tumor natural killer cell infiltration [43]. In contrast, Dubois-Vedrenne et al. proposed that increased tumor expression of chemerin led to impaired tumor angiogenesis and increased tumor necrosis and that the effect of chemerin on inflammatory cell influx was not a mechanism for its role in lung cancer [37]. In Lewis lung carcinoma cells engineered to express an active form of chemerin, tumor growth in vivo was decreased compared to non-transfected cells, and this was associated with a decrease in angiogenesis but no change in cellular recruitment to the tumor microenvironment [37]. Interestingly, similar results were seen if chemerin was only expressed in surrounding keratinocytes, and not in tumor cells, suggesting that microenvironment-specific, not just tumor-specific, levels of chemerin are important in regulating cancer growth and proliferation. Finally, when examining the role of chemerin in specific cell types of NSCLC, in a group of patients with NSCLC with adenocarcinoma, the expression of retinoic acid receptor responder 2 (RARRES2; chemerin) was identified as a marker for overall and progression-free survival in a seven-gene signature, with increased chemerin levels correlating with improved survival [44]. The authors propose that these findings were the result of a change in the tumor microenvironment with a decrease in the levels of M2 macrophage subtypes [44].

Several studies have examined chemerin signaling in lung cancer, with most of these studies focusing on the CMKLR1 receptor. In the study above by Dubois-Vedrenne et al., the effects of tumor-expressed chemerin in their model were shown to be dependent on CMKLR1 expression [37]. Similar results were seen when examining the tumor expression of CMKLR1 in lung adenocarcinoma, wherein the tumor expression of CMKLR1 was found to be negatively correlated with survival [45]. In addition, the expression of CMKLR1 positively correlated with the presence of tumor-infiltrating lymphocytes (TILs), with higher numbers of TILs in tumors with higher CMKLR1 expression, thereby providing a potential mechanism for the improved survival in the study [45]. The increased expression of CMKLR1 has also been shown in tumor-associated macrophages, again suggesting a role for CMKLR1 in regulating tumor growth [46]. It is unknown at present if a similar finding is seen in other cellular types of NSCLC beyond adenocarcinoma. In contrast to CMKLR1, a role of chemerin signaling through GPR1 in lung cancer has not yet been explored. However, the role of CCRL2 is beginning to be examined. The expression of the CCRL2 receptor appears to inhibit tumor growth, as a deficiency in CCRL2 expression promoted cancer growth in both p53- and urethane-induced lung cancer models [47]. Mechanistically, this was thought to be mediated by the decrease in inflammatory cell recruitment, particularly NK cells [47]. The above studies examined the role of chemerin in NSCLC; however, at present, no studies examine serum or tissue levels of chemerin in patients with small-cell lung cancer, and only a single study examines its role in other forms of lung cancer. The expression of RARRES2 was found to be increased in mesothelioma cells compared to normal mesothelial cells, suggesting a role for chemerin in pleural mesothelioma [48].

## 7. Emerging Role of Chemerin in Interstitial Lung Disease

Interstitial lung disease (ILD) constitutes a large collection of diseases affecting the interstitium of the lung, causing inflammation and resulting in lung fibrosis. Symptoms of patients include cough, shortness of breath, and exercise intolerance and may be associated with oxygen desaturation and the need for supplemental oxygen. These diseases can be caused by autoimmune processes, medications, environmental exposures, and chronic responses to infection, or have an unknown cause and be labeled as idiopathic. Common interstitial lung diseases include hypersensitivity pneumonitis, non-specific organizing pneumonia, giant-cell interstitial pneumonia, organizing pneumonia, and usual interstitial pneumonitis (clinically known as idiopathic pulmonary fibrosis). Several excellent review articles have recently been published reviewing research on ILD [49,50]. A potential role of chemerin in interstitial lung disease is only beginning to be examined. Recently, the role of chemerin in systemic sclerosis has been explored. Systemic sclerosis is an autoimmune disease with several systemic effects including fibrosis of the skin and esophageal dysmotility. More worrisome features, and those more directly related to mortality and overall outcomes, are those associated with the pulmonary complications of pulmonary arterial hypertension and interstitial lung disease. Although levels of serum chemerin have varied across studies of patients with systemic sclerosis ([51,52]), with results depending on duration and disease involvement (diffuse versus limited), no consistent findings of differences in chemerin expression have been identified in patients with systemic sclerosis and concomitant interstitial lung disease and those without ILD. In a study by Sanges et al., chemerin levels were similar in their cohort of patients with SSc with extensive ILD compared to healthy controls [53]. Similarly, Chighizola et al. showed that chemerin levels in their patient with SSc with ILD also did not differ compared to controls. However, several points should be highlighted [51]. First, in the study by Chighizola et al., serum chemerin levels were decreased in patients with SSc with diffuse disease and correlated with the duration of disease, with lower levels found in earlier stages of disease [51]. The researchers hypothesize that this may be due to the increased fibrosis seen in earlier stages of disease associated with increased expression of the pro-fibrotic mediator transforming growth factor alpha, which downregulates the expression of peroxisome proliferative-activated receptor gamma, a regulator of chemerin expression. Accordingly, chemerin levels may be altered in the earlier stages of SSc-associated ILD and be a biomarker of disease. Second, the duration or severity of SSc-associated ILD in the patient cohort was not included in any subgroup analysis. In contrast to SSc-associated ILD, more consistent findings were observed in these studies of an increased serum chemerin level in patients with SSc with pulmonary arterial hypertension. Taken together, additional studies are necessary to assess the potential role of chemerin in SSc-associated ILD and in ILD seen in other forms of connective tissue disease.

A potential role of chemerin in idiopathic pulmonary fibrosis has also begun to be explored in several early studies. The levels of chemerin in the serum were significantly higher in patients with IPF compared to healthy controls [54]. Similar findings were seen by Lavis et al., wherein chemerin levels were increased in the serum and bronchoalveolar lavage fluid from patients with IPF compared to samples obtained from healthy controls [55]. Although further mechanisms were not explored in these studies, the results are suggestive of a role of chemerin in the clinical manifestations of IPF. To further explore these findings, Lavis et al. examined the role of chemerin in a bleomycin model of ILD using mice deficient in chemerin expression. Their findings showed that mice deficient in chemerin expression developed significantly less lung fibrosis than control mice, and this correlated with a decrease in immune cell recruitment to the lung after exposure to bleomycin (Lavis, et al.). These findings were also associated with a decrease in fibroblast activation protein (FAP) in bronchoalveolar fluid in chemerin-deficient mice. Finally, Zielinski et al. also observed an increase in serum levels of chemerin in patients with sarcoidosis, an interstitial lung disease without a known etiology [54]. In regard to chemerin signaling, a study examining lung fibrosis after bleomycin exposure using CMKLR1-deficient mice found that the lack of CKMLR1 expression resulted in an increase in neutrophil, but a decrease in NK cell, recruitment to the lung as well as an increase in lung fibrosis [56]. In contrast, however, in another study, an increase in CMKLR1 levels in the BAL fluid of patients with IPF was associated with an overall worse prognosis [57]. Similar findings were observed when examining CMKRL1 expression in lung macrophages from bleomycin-exposed mice. Altogether, further studies are needed to better understand the role of chemerin in regulating the pathophysiology of pulmonary fibrosis. This likely will entail examination of chemerin expression levels and signaling in fibroblasts and alveolar epithelial cells, as well as the lung endothelium.

## 8. Preliminary Findings in Other Lung Diseases

There have been a few studies that have begun to explore the role of chemerin in cystic fibrosis and tuberculosis, but no inference can be made at present regarding the role of chemerin in these diseases. For tuberculosis, a role for chemerin could be expected due to the importance of macrophages in controlling the disease [58]. In a study examining this question, the levels of chemerin were found to be lower in patients with tuberculosis compared to controls [58]. However, these findings—per the authors—were thought to be due to the significant differences in body weight between the two groups, with patients with tuberculosis having a significantly decreased body mass index (BMI). As it relates to cystic fibrosis, in a study examining chemerin levels in children, there was no difference in serum chemerin levels between those with cystic fibrosis and controls [59]. In addition, there was no correlation between chemerin levels and nutritional status or serum cytokine levels in the study. These results are somewhat surprising as chemerin is known to be involved with inflammation, particularly during infections, which are commonly present in patients with cystic fibrosis. In contrast, in a study by Machurajet et al., serum chemerin levels were significantly higher in patients with cystic fibrosis compared to controls, with these values correlating with levels of C-reactive protein (CRP) [60]. The authors suggest that chemerin may correlate with the severity of cystic fibrosis, but this was only based on the levels of CRP without examining for correlations of chemerin with lung function testing or a history of infections. Studies in adults with chronic ongoing infections and the presence of significant airflow obstruction may illustrate a role of chemerin in cystic fibrosis.

## 9. Conclusions

Chemerin is associated with a complex biology regarding its proteolytic cleavage, resulting in several unique peptides with their own activation status, site of generation, and role in regulating cell migration and cytokine release. Although much is known regarding the cleavage sites that activate chemerin, less is known regarding the specific sites of activation and the overall level and activity of the constituents in tissues and at the cellular level. An improved understanding of these principles is required before the targeting of chemerin in the treatment of specific lung disease (Figure 2). The studies presented herein reveal that chemerin plays a role in airway disease, including asthma and COPD, but its specific role may depend on the airway irritant or allergen that is inducing the airway disease. Chemerin has also been shown to be involved in regulating the response to infection in the lung, particularly after viral infection, although a comprehensive evaluation of chemerin in respiratory infection with a wide variety of pathogens remains to be fully conducted. The use of additional animal models and genetically deficient mice lacking a response to chemerin will assist in our understanding of the role of chemerin in lung infections. In regard to interstitial lung disease, small studies suggest a role of chemerin in idiopathic pulmonary fibrosis, but larger, more extensive studies are needed. Finally, chemerin appears to regulate lung cancer growth and progression through several mechanisms including immune cell recruitment and angiogenesis. In lung cancer, both tumor expression and peri-tumor levels are likely important to the pathophysiology of lung cancer.

## Figures and Tables

**Figure 1 biomedicines-13-01354-f001:**
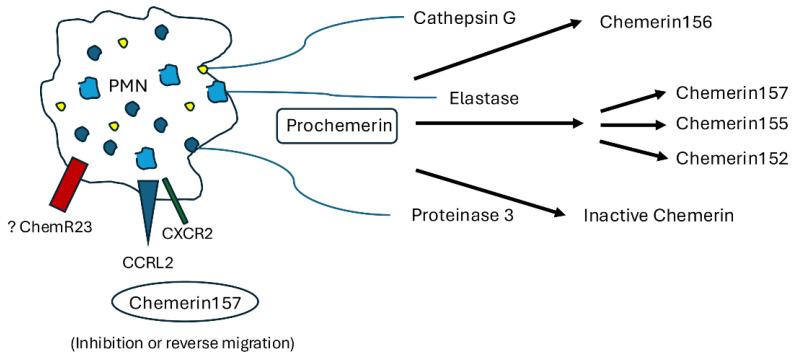
Neutrophil-mediated proteolytic cleavage of prochemerin and chemerin-induced neutrophil activation.

**Figure 2 biomedicines-13-01354-f002:**
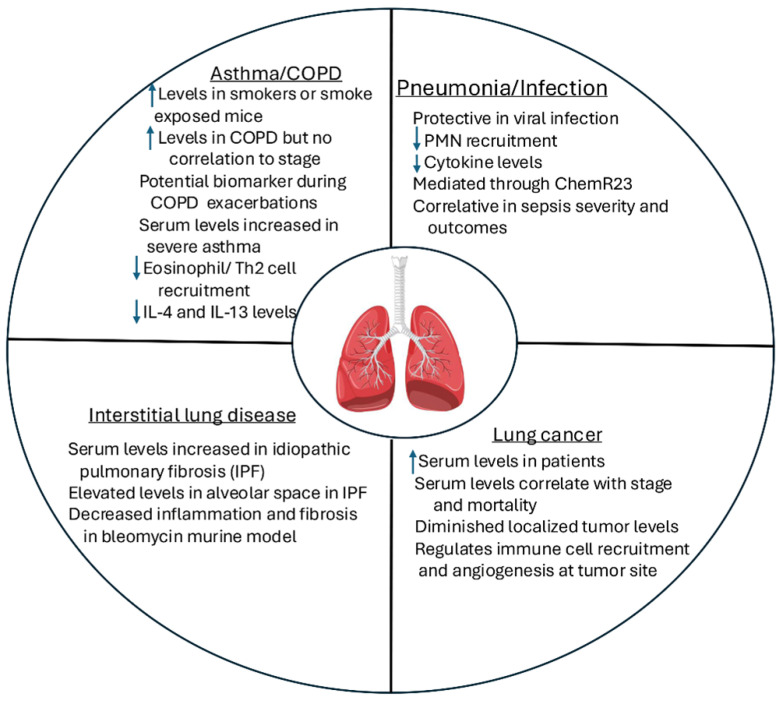
Role of chemerin in lung disease.

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
