# Peer review of "The Role of Chemerin in Neutrophil Activation and Diseases of the Lung"

_biomedicines, 2025, doi:10.3390/biomedicines13061354_

Round 1
Reviewer 1 Report
Comments and Suggestions for Authors
Dear authors
The current manuscript aims to discuss the role of chemerin in neutrophil activation and examine what is currently known about its effect on lung diseases.
However, I have many comments:
The paragraphs are very long and contain contradicting facts, such as in the discussion of research, not a review article. Thus, the points seem to be vague
Many facts lack cited references.
Many references are old, and only 4 are for 2023 or 2024.
The English language needs editing.
This review needs to be shorter and linked together, some paragraphs are more than 30 lines
Other comments in the attached manuscript
Best regards,

Dear Editor
The current manuscript aims to discuss the role of chemerin in neutrophil activation and examine what is currently known about its effect on lung diseases.
However, I have many comments:
The paragraphs are very long and contain contradicting facts, such as in the discussion of research, not a review article. Thus, the points seem to be vague
Many facts lack cited references.
Many references are old, and only 4 are for 2023 or 2024.
The English language needs editing.
This review needs to be shorter and linked together, some paragraphs are more than 30 lines
Other comments in the attached manuscript
Best regards,
Author Response
I thank the reviewer for their thoughtful review of our manuscript. The overall goal of this review was to examine what is known regarding the role of chemerin in lung disease. I agree with the reviewer that the specific role of chemerin in an individual lung disease does appear to be contradictory but hat is what can be currently gleamed from the relatively few studies undertaken for each specific lung disease. Conflicting results from each study are likely due to the various animal models utilized and because of the complex biochemistry of chemerin itself.
In undertaking this reviewer article an extensive literature search was undertaken up to March 2025. All articles reviewed for the manuscript were those identified during this search with cross references examined from these identified publications. The fact that only a small portion of the references were from 2023 and 2024 highlights the recent paucity of research involving chemerin in specific lung diseases and illustrates the unmet need for future research.
Reviewer 2 Report
Comments and Suggestions for Authors
The manuscript entitled "The role of chemerin in neutrophil activation and lung diseases" aims to elucidate the function of this adipokine in neutrophil activation and its involvement in various lung diseases such as asthma, COPD and cancer. The review is interesting, well-organized and well-written. However, I have a few suggestions that I believe could enhance the overall quality and impact of the manuscript.
1. It would be very informative to include two figures: one illustrating the molecular mechanisms of chemerin and another depicting its association with different lung diseases.
2. If available in the literature, it would be beneficial to discuss the association of chemerin with other lung conditions such as cystic fibrosis, tuberculosis and asbestosis.
Author Response
I thank the reviewer for their careful review of the manuscript and for their helpful suggestions for improving this review article. I agree with the reviewer that figures showing the biochemistry of chemerin and its role in lung disease would be helpful to disseminate the findings presented in the manuscript. I have now included these figures in the revised version if the manuscript. In preparing the manuscript, I performed an extensive literature search for the role of chemerin in a broad array of pulmonary diseases and have presented examples with at least several studies published discussing a role for chemerin in these diseases. I agree with the reviewer that it would be interesting to understand the role of chemerin in other pulmonary diseases including tuberculosis, cystic fibrosis, and after asbestos exposure resulting in pleural plaques or asbestosis. This is particularly true for cystic fibrosis which is associated with bacterial infections, for which chemerin has been shown to play a role. For these diseases only limited studies have been published. In the revised manuscript I have now included what is known of a potential role of chemerin in cystic fibrosis and tuberculosis. Unfortunately, no information of the role of chemerin in asbestos induced lung disease could be identified.
Reviewer 3 Report
Comments and Suggestions for Authors
The paper systematically reviews the role of chemerin in neutrophil activation and lung diseases, providing a rich analysis of the biochemical mechanisms involved. By integrating multiple studies, it reveals the dual role of chemerin in various lung diseases, such as asthma, chronic obstructive pulmonary disease (COPD), and lung cancer, offering important clues for future research directions. The paper emphasizes the complexity of chemerin and its multifaceted roles in inflammatory responses, presenting potential targets for the treatment of related diseases. This review is comprehensive, and the research content is significant, making it a good review article. I have the following questions:
1.The paper should include a model diagram illustrating the role of chemerin in neutrophil activation and lung diseases to help readers quickly grasp the main content of the review.
2.Can the authors provide some reports on the effects of drugs for treating lung diseases on chemerin to illustrate the relationship between lung disease treatment and chemerin?
3.The paper needs a concluding paragraph summarizing the entire text.
Author Response
Reviewer 3
I thank the reviewer for their comprehensive review of the manuscript and for their insightful suggestions to improve the quality and impact of the manuscript. I agree with the reviewer that figures showing the effect of chemerin on neutrophil activation and its role in lung disease would be helpful to disseminate the findings presented in the manuscript and present clarity. I have now included these figures in the revised version if the manuscript. I agree with the reviewer that information of the effect of medications used to treat pulmonary disease on the levels or activity of chemerin would be beneficial and would advance our understanding of the role of chemerin in acute inflammation and provide future targets in drug design. During the initial literature review for this manuscript I did not identify any studies to answer these specific questions and did not identify any studies to directly answer this question on a subsequent literature review. Finally, I have now included a summary paragraph in the revised version of the manuscript.
Round 2
Reviewer 1 Report
Comments and Suggestions for Authors
Dear authors
Please revise and correct your manuscript
Best regards
Reviewer 2 Report
Comments and Suggestions for Authors
I truly appreciate the efforts of the author to improve the manuscript. I think it is now suitable for publication.
Reviewer 3 Report
Comments and Suggestions for Authors
All my concerns have been addressed by author.